# Diverse prey capture strategies in teleost larvae

Duncan S Mearns[†‡], Sydney A Hunt[†§], Martin W Schneider, Ash V Parker, Manuel Stemmer, Herwig Baier*

Max Planck Institute for Biological Intelligence, Department Genes – Circuits – Behavior, Martinsried, Germany

## eLife Assessment

This **important** body of work uses state-of-the-art quantitative methods to characterize and compare behaviors across five different fish species to understand which features are conserved and which ones are differentiated. The **convincing** results from this study will be of interest to ethologists and also have potential utility in understanding the neural mechanisms leading to these behaviors.

*For correspondence:
herwig.baier@bi.mpg.de

[†]These authors contributed equally to this work

Present address: [‡]Princeton Neuroscience Institute, Princeton, United States; [§]Max Planck Institute of Animal Behavior, Konstanz, Germany

Competing interest: The authors declare that no competing interests exist.

**Abstract** Animal behavior is adapted to the sensory environment in which it evolved, while also being constrained by physical limits, evolutionary history, and developmental trajectories. The hunting behavior of larval zebrafish (*Danio rerio*), a cyprinid native to streams in Eastern India, has been well characterized. However, it is unknown if the complement and sequence of movements employed during prey capture by zebrafish is universal across freshwater teleosts. Here, we explore the syntax of prey capture behavior in larval fish belonging to the clade *Percomorpha*, whose last common ancestor with cyprinids lived ~240 mya. We compared the behavior of four cichlid species from Lake Tanganyika endemic to deep benthic parts of the lake (*Lepidiolamprologus attenuatus*, *Lamprologus ocellatus*, and *Neolamprologus multifasciatus*) or inhabiting rivers (*Astatotilapia burtoni*) with that of medaka (*Oryzias latipes*), a fish found in rice paddies in East Asia. Using high-speed videography and neural networks, we tracked eye movements and extracted swim kinematics during hunting from larvae of these five species. Notably, we found that the repertoire of hunting movements of cichlids is broader than that of zebrafish, but shares basic features, such as eye convergence, positioning of prey centrally in the binocular visual field, and discrete prey capture bouts, including two kinds of capture strikes. In contrast, medaka swim continuously, track the prey monocularly without eye convergence, and position prey laterally before capturing them with a side swing. This configuration of kinematic motifs suggests that medaka may judge distance to prey predominantly by motion parallax, while cichlids and zebrafish may mainly use binocular visual cues. Together, our study documents the diversification of locomotor and oculomotor adaptations among hunting teleost larvae.

## Introduction

In recent decades, biologists have increasingly relied on a handful of genetically tractable species to study questions related to behavioral mechanisms and their underlying neural circuitry (*Devineni and Scaplen, 2021*; *Piggott et al., 2011*; *Zhu and Goodhill, 2023*). However, it is often not clear to what extent these findings in model organisms can be generalized to other taxa or even closely related species. For example, recent comparative behavioral studies in drosophilids that diverged less than 40 mya found differences in both locomotion (*York et al., 2022*) and sequencing of spontaneous behaviors (*Hernández et al., 2021*). For vertebrates, the zebrafish larva has become a dominant model for

understanding the neuronal circuits and pathways controlling innate behavior (*Baier and Scott, 2009*; *Friedrich et al., 2010*; *Gahtan and Baier, 2004*; *Orger and de Polavieja, 2017*; *Portugues and Engert, 2011*). While many aspects of the visuomotor transformations and underlying neural circuitry for prey capture have been revealed in this species (reviewed by *Baier and Scott, 2024*; *Zhu and Goodhill, 2023*), it is not known if this is the general solution for larval teleosts or a derived adaptation to the zebrafish's specific ecological niche.

Here, we applied advances in computational ethology, such as automated tracking with deep neural networks (*Mathis et al., 2018*; *Pereira et al., 2019*) and unsupervised analyses (*Berman et al., 2014*; *Marques et al., 2018*; *Mearns et al., 2020*; *Wiltschko et al., 2015*) to compare hunting behavior of zebrafish to the Japanese rice fish, medaka (*Oryzias latipes* [OL]) (*Mano and Tanaka, 2012*) and four cichlids from Lake Tanganyika (*El Taher et al., 2021*; *Higham et al., 2007*). Medaka and cichlids belong to a diverse clade of teleosts known as percomorphs, whose last common ancestor with ostariophysi, such as zebrafish and Mexican cavefish, lived ~240 mya (*Betancur-R et al., 2017*; *Figure 1A*). While medaka and cichlids are only distantly related (~100 mya; *Betancur-R et al., 2017*), the haplochromine species and the three lamprologine species studied here diverged recently, within the past 3 million years (*Ronco et al., 2021*). We discovered that swim kinematics and the use of eye movements differ qualitatively and quantitatively between these species. This divergence may be driven, in part, by differences in the sensorineural mechanisms underlying prey detection. We believe that a detailed and quantitative description of behavior is an important first step toward uncovering underlying neural mechanisms and relating behavioral differences between species to their ecological niches.

## Results

### Artificial neural networks track swimming behavior and eye movements in percomorph larvae

At 5–7 days post-fertilization (dpf), zebrafish larvae robustly feed on small prey items such as paramecia. As juveniles, they feed on zooplankton such as brine shrimp (artemia), which are too large for larvae to ingest. To stage-match larvae of different species to zebrafish, we first identified when they started feeding (*Figure 1B*). Cichlid larvae started feeding later than zebrafish, from 12 to 14 dpf, and medaka (OL) started feeding at approximately 10 dpf. We next adapted an experimental paradigm used to study prey capture in zebrafish to these other species (*Mearns et al., 2020*). Individual larvae were placed in chambers with prey items (either artemia or paramecia). We recorded each animal for 15 min using a high-speed camera (*Figure 1—video 1* through *Figure 1—video 6*; see Materials and methods). We then used neural networks to extract tail pose, eye movements, and prey locations from videos (*Figure 1C*). We trained a 12-point SLEAP (Social Leap Estimates Animal Poses) model (*Pereira et al., 2022*) to track the tail and a 7-point model to track the eyes of larvae (*Figure 1E and F*; *Figure 1—video 1*). We tracked prey position using YOLO (You Only Look Once) (*Redmon et al., 2016*; *Figure 1D*). We subsequently used these estimated pose dynamics (*Figure 2*) and prey location information to compare hunting and swimming across species.

### Larvae of different species have diverse swim patterns

At the first-feeding stage, zebrafish larvae swim in discrete, discontinuous bouts (*Budick and O'Malley, 2000*), which merge into continuous swimming at the juvenile stage (*Westphal and O'Malley, 2013*). In contrast, *Danionella cerebrum*, a close relative of zebrafish, exhibits slow continuous swimming as larvae (*Rajan et al., 2022*). It is not known how larval swimming patterns differ between more distantly related species.

We found marked variation in the continuity of swimming in early percomorph larvae (*Figure 1G*; *Figure 1—figure supplement 1*). On one extreme, medaka (OL) and *A. burtoni* (AB) swam with a continuous 'gliding' style characterized by sustained, uninterrupted tail undulations over many seconds with only short breaks between swimming episodes, similar to *Danionella*. On the other extreme, *L. ocellatus* (LO) and *N. multifasciatus* (NM) had an intermittent swimming style, often resting at the bottom of the chamber for minutes at a time with quiescent periods interrupted by short, rapid bursts of activity. *L. attenuatus* (LA) showed a behavior intermediate to these extremes, alternating between

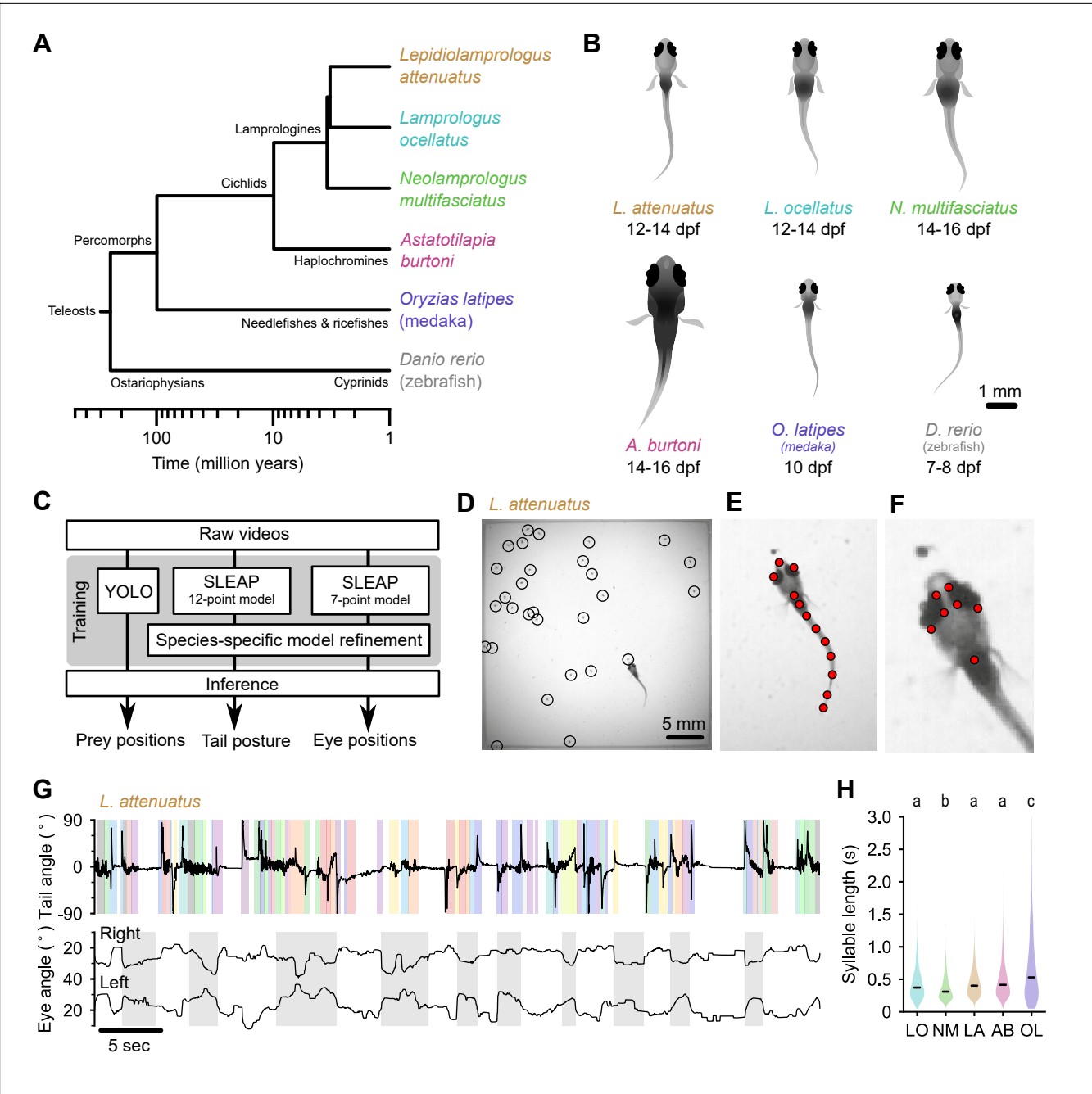

**Figure 1.** Tracking prey capture behavior in five species of fish larvae. (**A**) Phylogenetic relationship between percomorph species used in this study and their relationship to zebrafish (a cyprinid). Based on *Betancur-R et al., 2017*. (**B**) Schematics of larvae of each species studied here (and zebrafish, *Danio rerio*, for reference). Scale bar: 1 mm. (**C**) Outline of tracking procedure. Raw video frames were analyzed with three different neural networks to extract tail pose, eye pose, and prey position in each frame. Kinematic features were extracted from raw pose data and used for subsequent analyses. (**D–F**) Tracking output for each neural network for a single frame from a recording of a *Lepidiolamprologus attenuatus* larva, showing identified artemia (**D**), tail points (**E**), and eye points (**F**). (**G**) One minute of tail and eye tracking from *L. attenuatus* showing tail tip angles (top trace) and right and left eye angles (bottom traces) relative to midline. Bottom trace, gray shaded boxes: automatically identified prey capture periods, when the eyes are converged (see *Figure 3*). Top trace, shaded boxes: automatically identified and classified behavioral syllables. Color corresponds to cluster identity in *Figure 5*. (**H**) Distribution of syllable length across species for all fish combined. Black bars indicate median; letters denote significance groups (bootstrap test difference between medians with Bonferroni correction). See also *Figure 1—figure supplement 1*.

The online version of this article includes the following video and figure supplement(s) for figure 1:

**Figure supplement 1.** Diversity of swimming behavior in teleost larvae.

*Figure 1 continued on next page*

*Figure 1 continued*

**Figure 1—video 1.** Pose tracking in freely swimming *L. attenuatus*, slowed ×10.
https://elifesciences.org/articles/98347/figures#fig1video1

**Figure 1—video 2.** Example of *L. attenuatus* freely swimming and hunting artemia, played at half speed.
https://elifesciences.org/articles/98347/figures#fig1video2

**Figure 1—video 3.** Example of *L. ocellatus* freely swimming and hunting artemia, played at half speed.
https://elifesciences.org/articles/98347/figures#fig1video3

**Figure 1—video 4.** Example of *Neolamprologus multifasciatus* freely swimming and hunting artemia, played at half speed.
https://elifesciences.org/articles/98347/figures#fig1video4

**Figure 1—video 5.** Example of *Astatotilapia burtoni* freely swimming and hunting artemia, played at half speed.
https://elifesciences.org/articles/98347/figures#fig1video5

**Figure 1—video 6.** Example of medaka (*O. latipes*) freely swimming and hunting paramecia, played at half speed.
https://elifesciences.org/articles/98347/figures#fig1video6

rapid tail beating and quiescence, each lasting on the order of a few seconds, a discontinuous style similar to zebrafish.

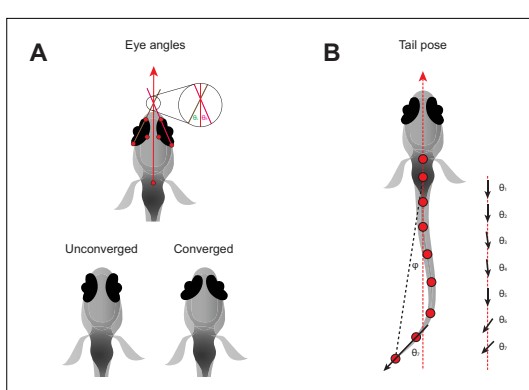

**Figure 2.** Pose information extracted from tracking data. (**A**) Definition of left ($\theta_L$) and right ($\theta_R$) eye vergence angles. The heading vector (red arrow) is the vector starting from the midpoint of the fins and passing through the midpoint of the eyes. The orientation of each eye is computed from the temporal and nasal points (long axis of the eye). The eye angle is the signed angle between the heading and long axis of the eye. Zero degrees indicates that the long axis of the eye is parallel with the heading. Nasalward rotation of the eye increases the vergence angle. Eye convergence angle is $\theta_L + \theta_R$. Bottom: schematic showing eyes in an unconverged and converged state. (**B**) Parameters used to estimate tail pose. Tail pose is measured starting from the middle of the swim bladder. We compute the vector between each consecutive pair of points along the tail. At each pair of points, we compute the signed angle between the corresponding vector and the midline of the fish (RHS: $\theta_{1-7}$, angles between black arrows and red dashed line), providing a seven-dimensional representation of the tail pose in each frame. For visualization purposes in *Figure 1G*, *Figure 1—figure supplement 1*, we plot the tail angle, which is the signed angle between the swim bladder and the final tail point relative to the heading (angle between dashed red and black lines, φ).

Analyses across the animal kingdom suggest that behavior is organized into sub-second kinematic motifs ('syllables'), even when movement is continuous (*Berman et al., 2014*; *Stephens et al., 2008*; *Tinbergen, 1951*; *Wiltschko et al., 2015*). Inspecting periods of swimming across percomorph species revealed significant substructure within swimming episodes (*Figure 1G*; *Figure 1—figure supplement 1*), suggesting that the continuous swimming of some species may in fact consist of kinematically discrete syllables fused together. We segmented continuous tail traces into discrete syllables using a change detection algorithm adapted from *Mearns et al., 2020* (see Materials and methods), which revealed changes in swim kinematics occurring on the order of tens to hundreds of milliseconds (median seconds: 0.33, LO; 0.29, NM; 0.38, LA; 0.36, AB; 0.49, OL). Furthermore, medaka (OL) exhibited considerably longer syllables than cichlids (p<0.01, bootstrap test difference between medians), while one cichlid species, NM, exhibited significantly shorter syllables than the others (p<0.05) (*Figure 1H*). These results highlight differences in how fish larvae pattern their swim episodes, which may relate to different sensorimotor strategies for prey detection and tracking.

## Larvae of different species use diverse strategies to hunt prey

Zebrafish larvae converge their eyes (nasalward rotation, *Figure 2A*) at the onset of hunting episodes and keep their eyes converged over the course of prey capture (*Bianco et al., 2011*; *Patterson et al., 2013*). Eye convergence aids prey capture by bringing prey into a binocular zone in the central visual field (*Gahtan et al., 2005*; *Gebhardt et al., 2019*). In contrast, some

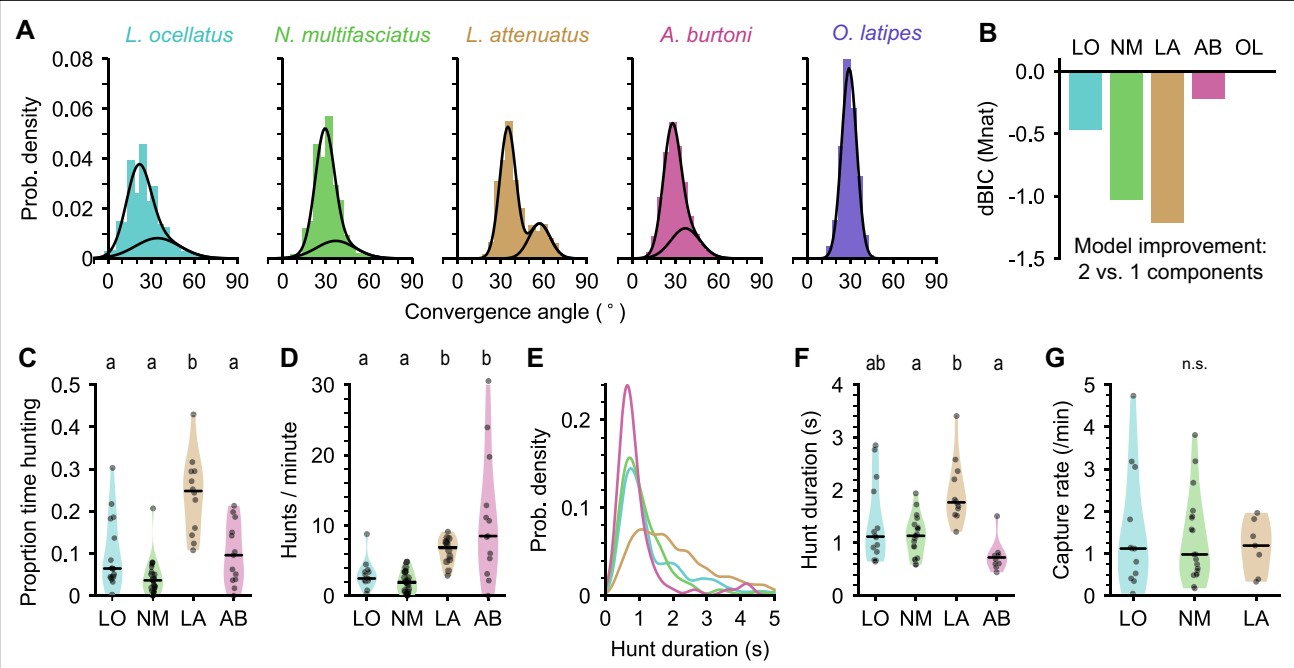

**Figure 3.** Eye movements during prey capture and statistics of hunting sequences. (**A**) Histograms of eye convergence angles for each species. The convergence angle is the angle between the long axes of the eyes. Shaded bars: normalized binned counts of convergence angles from all fish. Black lines: a best fit Gaussian mixture model for each species (with one or two components). For all cichlid species, the data are better modeled as being drawn from two underlying distributions. For medaka (*O. latipes* [OL]), the data are better modeled with a single underlying distribution. (**B**) Improvement in fit for a two-component over single-component Gaussian mixture model, assessed using Bayesian inference criterion (BIC). The BIC is a measure of model fit, while punishing over-fitting. Lower values are better. Two mixtures provide a better fit over a single mixture for all species except medaka (OL). (**C**) Proportion of time spent by cichlids engaged in prey capture within the first 5 min of being introduced to the behavior arena. Points are single animals; black bar is the median; letters indicate significance (different letters indicate difference between groups, α=0.05). (**D**) Hunting rate, measured as the number of times eye convergence is initiated per minute, within the first 5 min. Points are single animals; black bar is the median; letters indicate significance. (**E**) Kernel density estimation of hunt durations for each species (all animals pooled). *A. burtoni* (pink) hunts skew shorter, *L. attenuatus* (brown) hunts tend to be longer. (**F**) Median hunt duration for each animal compared across species. Black bar is the median across animals; letters indicate significance. (**G**) Capture rate (number of artemia consumed) per minute over the first 5 min for three cichlid species. Black bar is the median across animals. n.s., no statistically significant difference between groups.

The online version of this article includes the following figure supplement(s) for figure 3:

**Figure supplement 1.** Two-dimensional (2D) histograms of eye angles.

fish species such as the blind cave fish, *Astyanax mexicanus*, capture prey that are positioned laterally to their mouth (*Lloyd et al., 2018*; *Espinasa et al., 2023*). This shift in capture strategy appears to have happened over a relatively short timescale when cavefish diverged from their surface ancestors, and consequently, cavefish still converge their vestigial eyes even though they do not rely on vision for prey capture (*Espinasa and Lewis, 2023*).

We found that all four cichlid species examined here converged their eyes during prey capture, but medaka did not (*Figure 3A*; *Figure 1—video 2* through *Figure 1—video 6*). Two peaks in the distribution of eye convergence angles were visible in LA (*Figure 3A*, center), but other cichlids also spent significant time with their eyes converged more than 50°. From the raw videos, we could see that all cichlid species did converge their eyes during prey capture (e.g. *Figure 1—video 3* [LO], 00:06; *Figure 1—video 4* [NM], 00:22; *Figure 1—video 5* [AB], 00:40); however, this might not manifest as two peaks in the distribution of eye convergence angles if these species spent less time engaged in prey capture than LA and if variation in the exact placement of key points by SLEAP blurred the boundary between converged and unconverged states. To verify that two eye convergence states also existed in these species, but not in medaka (OL), we fitted Gaussian mixture models to the eye convergence data of each species with one or two components. For all species except OL, modeling the data using two underlying distributions provided a better fit than a single distribution (*Figure 3B*). A unimodal distribution of eye vergence angles in OL was also present when we plotted the joint

distribution of left and right eye angles (*Figure 3—figure supplement 1*), indicating that this species does not perform eye convergence, whereas for all cichlid species there are periods in the recordings when the eyes are converged.

Using eye convergence to identify hunting periods, we found that different cichlid species spent different amounts of time engaged in prey capture (*Figure 3C*). LO, NM, and AB spent less than 10% of the time hunting (median proportions: 0.064, LO; 0.036, NM; 0.096, AB). In contrast, LA, a piscivorous species as adults, spent about a quarter of the time actively engaged in prey capture, with all individuals tested spending at least 10% of their time and one individual spending over 40% of the time hunting (median: 0.25, p-values<0.05 comparing LA to all other species, Mann-Whitney U test with Bonferroni correction).

Surprisingly, although LO, NM, and AB all spent comparable total time engaged in prey capture, AB initiated hunts at a much higher rate, similar to LA (*Figure 3D*) (median hunts/min: 2.45, LO; 1.98, NM; 6.85, LA; 8.49, AB; p-values<0.05 comparing either LO or NM to LA or AB, Mann-Whitney U test with Bonferroni correction). AB hunts were also shorter than other species, while LA hunts tended to be longer (*Figure 3E and F*). LO and NM individual hunt durations were similar to each other and intermediate to the other species (medians across fish [s]: 1.12, LO; 1.14, NM; 1.77, LA; 0.73, AB). Despite these differences in time engaged in prey capture, all three species consumed prey at similar rates of approximately one artemia per minute (*Figure 3G*).

Taken together, our results highlight differences among closely related cichlids in hunting behavior. LA are the most persistent hunters, initiating prey capture often and spending a longer time engaged in the behavior once initiated (*Figure 1—video 2*). In contrast, AB prey capture dynamics are characterized by a high rate of short-duration hunting episodes (*Figure 1—video 5*). Both LA and AB prey capture behaviors are examples of an active hunting strategy. On the other hand, LO and NM initiate prey capture rarely, spending much time resting at the bottom of the chamber and moving only occasionally when they dart toward prey (*Figure 1—videos 3 and 4*), characteristic of a sit-and-wait predation strategy.

## Cichlids center prey within a strike zone

Zebrafish larvae strike at prey once it is localized in the center of their visual field and ~0.5 mm away (the 'strike zone') (*Mearns et al., 2020*; *Patterson et al., 2013*). This centering behavior is impaired when animals are blinded in one eye (*Mearns et al., 2020*). To test whether cichlids similarly center prey within a strike zone, we identified the most likely targeted artemia during each hunting episode and studied how the prey moved through the visual field as hunting sequences progressed (*Figure 4A*, see Materials and methods) (n events: 176, LO; 361, NM; 756, LA). This revealed that, within each prey capture sequence, prey became increasingly localized to the near-central visual field over time. Computing kernel density estimates of prey distributions at the beginning, middle, and end of each hunting episode revealed that cichlids initiate prey capture at a wide range of distances, and that they center prey in the visual field in the early stages of a hunting sequence (*Figure 4B*). For instance, the azimuthal angle of prey decreased over the start of the hunting episode in LO and NM (p<0.01, bootstrap test difference between medians) (*Figure 4D*), but not over the second half (p>0.05). Curiously, LA often initiated prey capture when artemia were already in the center of the visual field (*Figure 4A, B, and D*), which may be due to them restarting hunts on recently expelled prey (see description of spitting behavior below and *Figure 1—video 6*). On average, the cichlids' eyes converged when prey were ~5 mm away (median distance, mm: 5.43, LO; 3.98, NM; 4.17, LA). However, all species could detect prey up to 15 mm away (*Figure 4C*), and the distance to the prey decreases gradually over the course of the hunting episode, in contrast to centering, which tends to occur at the beginning.

In all cichlid species analyzed, prey were highly localized to a central strike zone ~1–2 mm away at the end of prey capture (*Figure 4B and C*). The tails of the distributions in *Figure 4C* are likely due to aborted hunting events. Together, these results demonstrate that cichlids are able to detect prey at a large distance and possess the fine sensorimotor control required to localize prey to a strike zone.

## Fish larvae of different species share a common pose space

Zebrafish larvae preferentially perform different kinds of swims during prey capture and exploration (*Borla et al., 2002*; *McElligott and O'Malley, 2005*). During prey capture, different swim types mediate distinct transformations of the visual scene: J-turns center prey in the visual field, approach

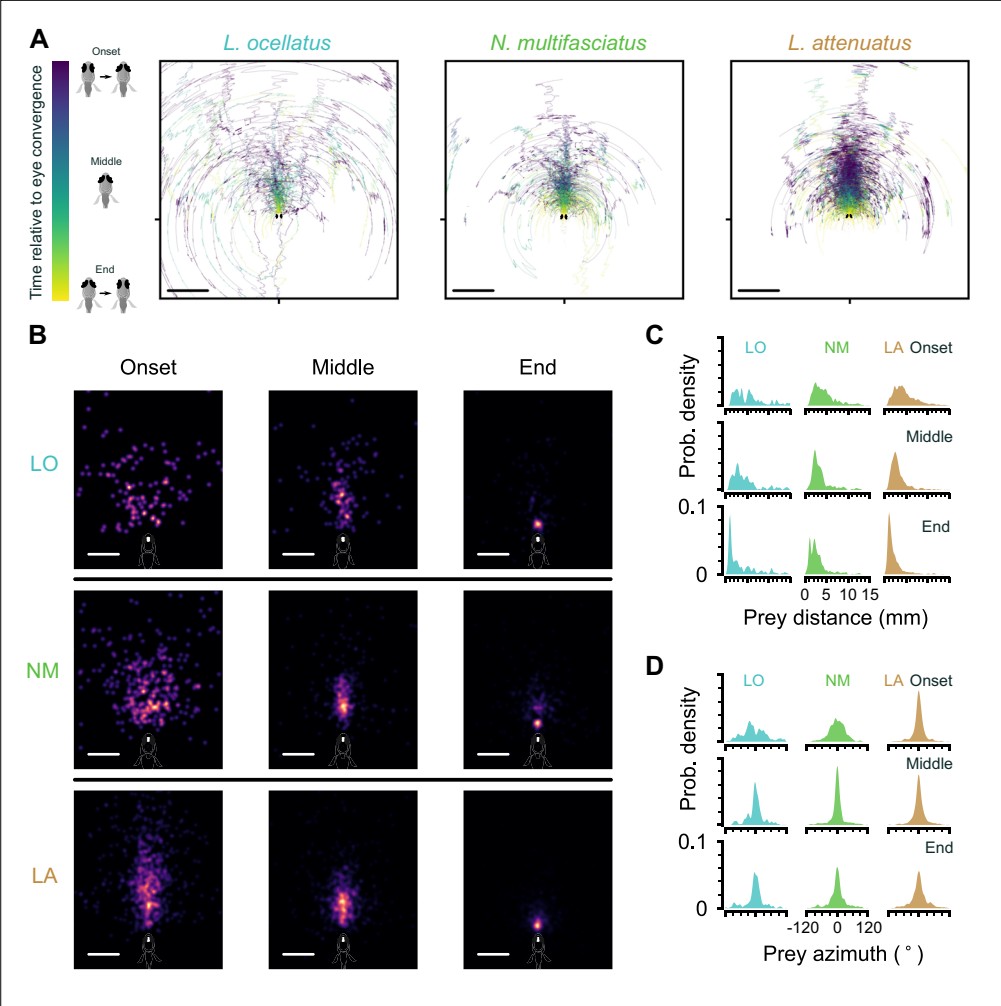

**Figure 4.** Location of prey in the visual field during prey capture in cichlids. (**A**) Trajectories of prey in the visual field for all automatically identified and tracked hunting events. Each prey is represented by a single line that changes in color from blue to yellow from the onset of eye convergence to when the eyes de-converge (end). Middle indicates the midpoint between eye convergence onset and end. Scale bar: 5 mm. (**B**) Kernel density estimation of the distribution of prey in the visual field across all hunting events. Rows: individual species. Columns: snapshots showing the distribution of hunted prey items at the beginning, middle, and end of hunting sequences. Scale bar: 2 mm. (**C, D**) Kernel density estimation of prey distance (**C**) and azimuthal angle from the midline (**D**) at the onset (top), in the middle (center), and at the end (bottom) of hunting episodes. Each column shows the distribution of all events for a single species.

swims bring prey to the strike zone, and strikes are deployed to capture prey. The identification and classification of these distinct swims have been aided by finding low-dimensional representations of behavior (*Johnson et al., 2020*; *Marques et al., 2018*; *Mearns et al., 2020*).

Due to neural and mechanical constraints, it is often possible to capture variation in tail shape over time with a greatly reduced number of dimensions, which serves to eliminate tracking noise and aid analysis (*Berman et al., 2014*; *Mearns et al., 2020*; *Stephens et al., 2008*; *York et al., 2022*). Performing principal components analysis (PCA) on all species combined revealed that tail pose dynamics are similarly low-dimensional (*Figure 5A*, black line; three principal components [PCs] explain >90% variance). These results also hold when each species is analyzed individually (*Figure 5A*, colored lines). As with zebrafish larvae (*Girdhar et al., 2015*; *Mearns et al., 2020*), the PCs correspond to a harmonic series (*Figure 5B*), with the first PC capturing 'turniness', and PCs 2 and 3 capturing tail oscillations during locomotion. The first three PCs were similar across species (*Figure 5C*), allowing swim kinematics to be compared across species.

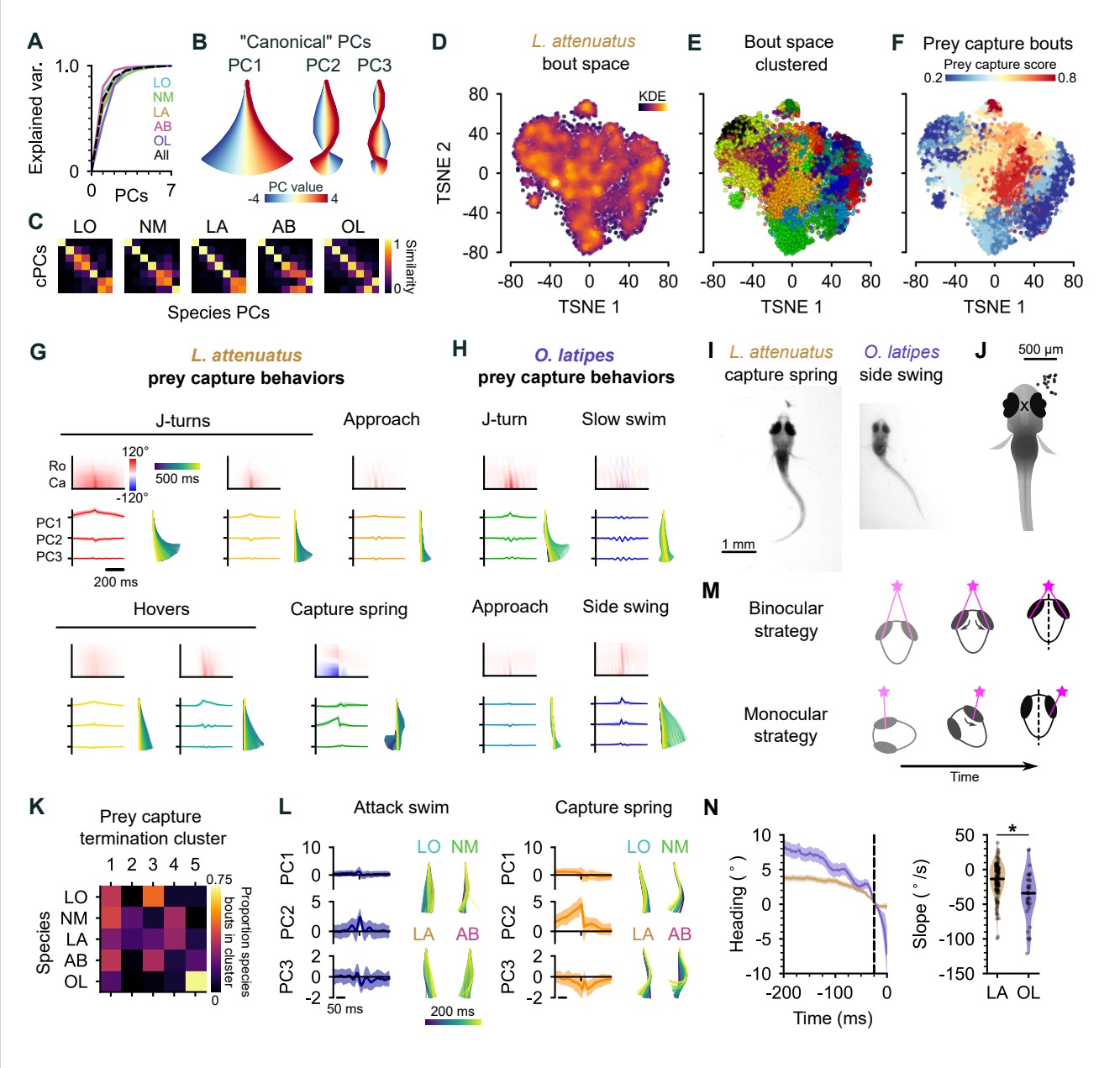

**Figure 5.** Interspecies comparison of prey capture strategies. (**A**) Cumulative explained variance for the 'canonical' principal components (PCs) obtained from all species (black dotted line) and PCs for each species individually (colored lines). In all cases, three PCs explain >90% of the variance in tail shape. (**B**) 'Eigenfish' of the first three canonical PCs. Each PC represents a vector of angles from the base to the tip of the tail (oriented with the fish facing up). At a given moment, the shape of the tail can be described as a linear combination of these vectors. Colors correspond to the tail shape obtained by scaling each PC from –4 to 4 standard deviations from the mean. (**C**) Each species' eigendecomposition compared against the canonical PCs computed for all species together. Color intensity represents the cosine similarity between pairs of vectors. The strong diagonal structure (particularly in the first three PCs) shows that similar PCs are obtained by analyzing species separately or together. (**D**) Behavioral space of *L. attenuatus* (LA). Each point represents a single bout. Bouts are projected onto the first three PCs, aligned to the peak distance from the origin in PC space and then projected into a two-dimensional space using t-distributed stochastic neighbor embedding (t-SNE). Color intensity represents the density of surrounding points in the embedding. (**E**) Clustered behavioral space of LA. Clusters (colors) are computed via affinity propagation independently of the embedding. (**F**) Prey capture and spontaneous bouts in LA. Prey capture score is the probability that the eyes are converged at the peak of each bout. Bouts are colored according to the mean prey capture score for their cluster. Blue: clusters of bouts that only occur during spontaneous swimming; red: clusters of bouts that only occur during prey capture. (**G, H**) Example prey capture clusters from LA (**G**) and medaka, *O. latipes* (OL) (**H**). For each cluster, top left: mean rostrocaudal bending of the tail over time; bottom left: time series of tail pose projected onto first three PCs (mean ± standard deviation); bottom right: reconstructed tail shape over time for mean bout. (**I**) Representative frames of an LA capture spring (left) and OL (medaka) side swing (right),

*Figure 5 continued on next page*

*Figure 5 continued*

highlighting differences in tail curvature between these behaviors. (**J**) Location of prey in the visual field (black dots) immediately prior to the onset of a side swing. All events mirrored to be on the right. X marks the midpoint of the eyes, aligned across trials. (**K**) Comparison of clustered hunt termination bouts from all species. Termination bouts from all species were sorted into five clusters based on their similarity. Rows show the proportion of bouts from each species that were assigned to each cluster (columns). Cichlid bouts are mixed among multiple clusters, while medaka bouts (OL) mostly sorted into a single cluster. (**L**) Capture strikes in cichlids. Representative examples of tail kinematics during attack swims (left) and capture springs (right) from each species, including time series of tail pose projected onto the first three PCs (mean ± standard deviation, all species combined) shown for each type of strike. (**M**) Two hypotheses for distance estimation make different predictions of how heading (black dotted line) changes over time as fish approach prey (pink star). Top: fish maintain prey in the central visual field and use binocular cues to judge distance. Bottom: fish 'spiral' in toward prey, using motion parallax to determine distance. Black arrows indicate motion of prey stimulus across the retina. (**N**) Change in heading over time leading up to a capture strike for LA (brown, n=113) and medaka (OL, purple, n=36). Left: time series of heading. Zero degrees represents the heading 25 ms prior to the peak of the strike (black dotted line). Mean ± s.e.m. across hunting events. Right: comparison of the rate of heading change, computed as the slope of a line fit to each hunting event 200–25 ms prior to the peak of the strike. The heading decreases more rapidly, leading up to a strike in medaka than in LA.

The online version of this article includes the following video and figure supplement(s) for figure 5:

**Figure supplement 1.** Behavioral spaces for each species.

**Figure 5—video 1.** *L. attenuatus* spitting behavior, played at half speed.

https://elifesciences.org/articles/98347/figures#fig5video1

**Figure 5—video 2.** *A. burtoni* swimming backward during prey capture, played at half speed.

https://elifesciences.org/articles/98347/figures#fig5video2

## Hunting sequences in cichlids share kinematic motifs with zebrafish but are more complex

A common low-dimensional pose space in drosophilids has aided with the identification of evolutionary trajectories in behavior (*York et al., 2022*). We therefore wanted to investigate the trajectories through pose space of cichlids and medaka (OL) to understand if they share common behaviors during hunting or if their behaviors have diverged over evolution.

For each species, we projected the time series of rostrocaudal tail angles for each distinct bout onto the first three PCs (*Mearns et al., 2020*). We then performed affinity propagation to identify clusters corresponding to different swim types and performed t-SNE to visualize these clusters in a low-dimensional embedding. LA spent the most time engaged in hunting in our experiments, and so we chose to focus on this species in particular when discussing the cichlid behavioral repertoire (*Figure 5D–G*); however, we performed similar analyses with all species (*Figure 5—figure supplement 1*), which revealed behavioral syllables shared across species (discussed below). We also found notable diversity in the number of syllables across species, ranging from 11 (LO) to 32 (AB), which is on par with or greater than the bout types that have been found in zebrafish larvae (*Marques et al., 2018*; *Mearns et al., 2020*). Multiple factors may contribute to this threefold difference in the number of behavioral syllables we identify between cichlid species. First, we note that species with the most clusters identified (29 and 32 in LA and AB, respectively) performed more hunting sequences than LO or NM (*Figure 3D*). If the highest variation in swim kinematics occurs during prey capture, we could be undersampling the behavioral repertoire of species such as LO. Similarly, we could be undersampling the diversity of spontaneous swims in species that spend a lot of time sitting on the bottom of the behavioral chamber. Alternatively, different cluster numbers could represent real variation in the behavioral repertoire of species, reflecting different hunting strategies. As noted above, LO and NM tend to hunt with a sit-and-wait strategy and, therefore, may perform fewer kinds of behavior syllables when darting toward prey. In contrast, AB and LA are much more active hunters and, therefore, may draw on a richer behavioral repertoire to orient toward, pursue, and switch between targets.

Similar to zebrafish, we found that cichlids have behavioral syllables that they preferentially perform during prey capture (*Figure 5F*). Some of these swims share characteristics with hunting bouts previously described in zebrafish, such as J-turns and approach swims, but we also identify new types that do not have clear analogs in zebrafish (*Figure 5G*). For example, cichlid larvae often 'hover' in place, oscillating their tail without any forward movement (*Figure 5G*). Such behaviors tend to be longer lasting than approach swims. We speculate that hovers may aid in keeping animals in place during hunting by preventing them from being moved by convection currents or by breaking their forward momentum following more rapid forward swims. Alternatively, they may allow animals to pause

between sensorimotor decisions during hunting sequences while keeping the tail motile, reducing the latency of the motor system to respond to descending motor commands. Interestingly, hovering is also present in some adult cichlids that use their lateral line to sense prey in the subsurface (*Schwalbe et al., 2016*). During hunting, cichlids will alternate between approaches and these hover swims.

Another feature of cichlid hunting behavior not present in zebrafish is tail coiling. Here, the tail coils into an S-shape over many hundreds of milliseconds leading up to a strike. The coil is released rapidly during the strike, allowing fish to spring toward the prey (*Figure 5G*, bottom center; *Figure 4I*; *Figure 1—video 2*, 00:12; *Figure 1—video 4*, 00:31). In addition to such capture springs, we have also observed suction captures (*Figure 1—video 3*, 00:33) and ram-like attack swims (*Figure 1—video 2* and 00:45) in cichlids.

Finally, we found that cichlids exhibit a range of poststrike maneuvers not present in zebrafish. These include expelling prey ('spitting') that has already been captured (*Figure 5—video 1*) and swimming backward to re-center prey in the visual field in cases where capture strikes miss (*Figure 5—video 2*). Spitting behavior often triggered another attempt to hunt and capture the same prey and was particularly common in LA. This behavior likely explains the high incidence of hunt initiations when prey items are already in the central visual field in this species (*Figure 4A, B, and D*). The function of spitting is not clear; it might indicate that the prey are slightly too large for LA to swallow. We also find that cichlid captures often occur in two phases, with the prey first being caught by the pharyngeal jaws (a second, internal set of jaws present in cichlids; *Liem, 1973*) before being ingested. Together, these results reveal a richer behavioral repertoire for pursuing, capturing, and reorienting toward missed prey in cichlids than in zebrafish.

## Cichlids share two common capture strikes that are distinct from medaka side-swing strikes

Studies in invertebrates have demonstrated that behavior can evolve through changing the kinematics of a behavior performed in a given sensory context (*Ding et al., 2019*), changing which sensory cues drive behavior (*Seeholzer et al., 2018*), or changing the transition frequencies between a common set of kinematic motifs (*Hernández et al., 2021*). It is not known to what extent these factors play a role in the evolution of vertebrate behaviors, nor the evolutionary timescales over which behaviors diverge to the point of becoming distinct. To address this question, we focused on the capture strike kinematics of cichlids and medaka (OL). The capture strike of zebrafish larvae has been extensively characterized, consisting of two distinct types: the attack swim and the S-strike. These strikes are deployed variably, dependent on experience and different prey distances (*Lagogiannis et al., 2020*; *McClenahan et al., 2012*; *Mearns et al., 2020*; *Patterson et al., 2013*). At the week-old stage, attack swims are the most common capture maneuver and consist of a symmetric oscillation of the tail that resembles spontaneous forward swims. During an S-strike, the tail does not oscillate, but rather bends into an S-shape and then straightens rapidly, causing larvae to lunge toward prey.

Medaka (OL) swimming is nearly continuous, and their eyes do not converge during prey capture. Nevertheless, they may still perform different behaviors during spontaneous swimming and prey capture. Investigating clusters of behavior in OL revealed types of swim that are similar to hunting bouts of cichlids and zebrafish. These included J-turns, approach swims, and slow swims (*Figure 5H*). The presence of these swims suggests that OL, like other species studied, may also have a suite of behaviors they preferentially perform during prey capture. Remarkably, we found that OL did not strike at prey in the central visual but rather captured prey in the lateral visual field with a side-swing behavior (*Figure 5H–J*). Although the eyes do not converge, there is sometimes a slight nasalward rotation of the ipsilateral eye leading up to and during the side swing (*Figure 5I*).

To test whether the side-swing behavior was unique to medaka (OL), we clustered capture strikes of cichlids and OL after projecting their tail kinematics into the common pose space, setting the number of clusters to the number of species (*Figure 5A*). We reasoned that if bouts were similar within each species but different between species, cluster identity would correlate with species identity. In contrast, if species shared similar kinematic motifs, clusters would contain bouts from all species. We found that strikes from all cichlid species were similar to each other, while the side-swing strike was highly enriched in OL (*Figure 5K*).

This analysis uncovered at least two kinematically distinct capture strikes in cichlids. The slower of these behaviors is similar to the zebrafish attack swim, characterized by a symmetric undulation of

the tail about the midline which propels the fish toward the prey, while the faster corresponds to the capture spring. The S-shape of the tail leading up to the capture spring shares structural similarities with the S-strike of zebrafish larvae (*Mearns et al., 2020*), differing primarily in the time course over which they occur. The S-strike of zebrafish may represent an accelerated form of the capture spring, forming and releasing over tens of milliseconds rather than hundreds of milliseconds seen in cichlids or, vice versa, the capture spring may represent an augmentation of the S-strike.

## Different prey capture strategies may reflect different sensorineural solutions to prey detection

Centralized prey and converged eyes in cichlids suggest that the mechanism they use to determine the distance to prey could be similar to that of zebrafish. One possibility is that these species use binocular disparity to judge depth (*Qian, 1997*). In contrast, we hypothesized that medaka (OL) might use a different, monocular strategy to judge the distance to prey, such as motion parallax (*Yoonessi and Baker, 2011*). These two mechanisms of depth perception make different predictions about how larvae should approach prey to make best use of visual cues (*Figure 5M*). To investigate these possibilities, we computed the change in heading leading up to a strike as a proxy for the change in visual angle of the prey for cichlids and medaka (*Figure 5N*). We found that the rate of change in heading was greater in medaka than in cichlids (median °/s: –13.4, LA; –34.1, OL; $p < 0.001$, Mann-Whitney U-test). These results are consistent with cichlids using binocular cues, such as binocular disparity, to determine distance to prey, while medaka may employ a monocular strategy.

## Discussion

Here, we have provided the first description of the hunting behavior of larval cichlids and medaka, popular model systems for understanding the neural and genetic basis of behavior (*Shima and Mitani, 2004*; *Jordan et al., 2021*; *Johnson et al., 2023*). We have found that cichlids share the same general hunting strategy to zebrafish: striking at prey localized to a binocular strike zone while the eyes are converged, with two distinct capture bout types whose kinematics are remarkably similar given these species diverged 240 mya (see *Figure 1A*). In contrast, medaka, which are phylogenetically intermediate to zebrafish and cichlids, deploy a different hunting strategy: striking at prey laterally in the monocular visual field with a side-swing behavior. *Figure 6* provides a schematic summary of similarities and differences among the teleost larvae investigated. Characterization of prey capture behavior in a wider range of species could reveal whether the similarities between zebrafish and cichlids represent conservation of an ancestral hunting strategy or convergent evolution.

What sensorineural mechanisms might underlie these two different hunting strategies? The side swing of medaka is reminiscent of hunting in (adult) blind cavefish (*Lloyd et al., 2018*), which position prey laterally prior to strikes using their mechanosensory lateral line. Adult cichlids are known to use their lateral line for prey capture (*Schwalbe et al., 2012*), and some percomorph species exhibit both an S-strike-like and a sideways capture strategy as adults (*New, 2002*). Therefore, it is conceivable that medaka larvae do not rely on vision to the same extent for hunting, but use their lateral line instead. However, lateral-line use is not uncoupled from eye convergence across the teleost clade. Zebrafish larvae can still hunt in the dark, as do blind zebrafish *lakritz* mutants, albeit with greatly reduced efficacy, and with their eyes converged (*Gahtan et al., 2005*; *Mearns et al., 2020*; *Patterson et al., 2013*). Larval cavefish, which evolved from sighted surface ancestors and are completely blind, exhibit vestigial eye convergence movements during prey capture before their eyes fully degenerate (*Espinasa et al., 2023*; *Espinasa and Lewis, 2023*). This comparative evidence suggests that the teleost brain employs cues from both sensory modalities, if available, to locate prey, but that eye movements are a poor predictor of the dominance of vision.

On the other hand, species that attack centrally located prey could be using different visual cues to judge prey distance than those that use a side swing to ingest food: binocular disparity (*Qian, 1997*) vs. motion parallax (*Yoonessi and Baker, 2011*). Zebrafish use binocular information (*Gahtan et al., 2005*; *Gebhardt et al., 2019*; *Henriques et al., 2019*) and stationary differences such as brightness and contrast (*Khan et al., 2023*) to estimate depth and distance. Here, we have shown that cichlid larvae also likely use binocular cues, while medaka's approach to prey conforms to a monocular strategy (see *Figure 5M and N*). Strikingly, cichlid and zebrafish larvae swim in bouts with

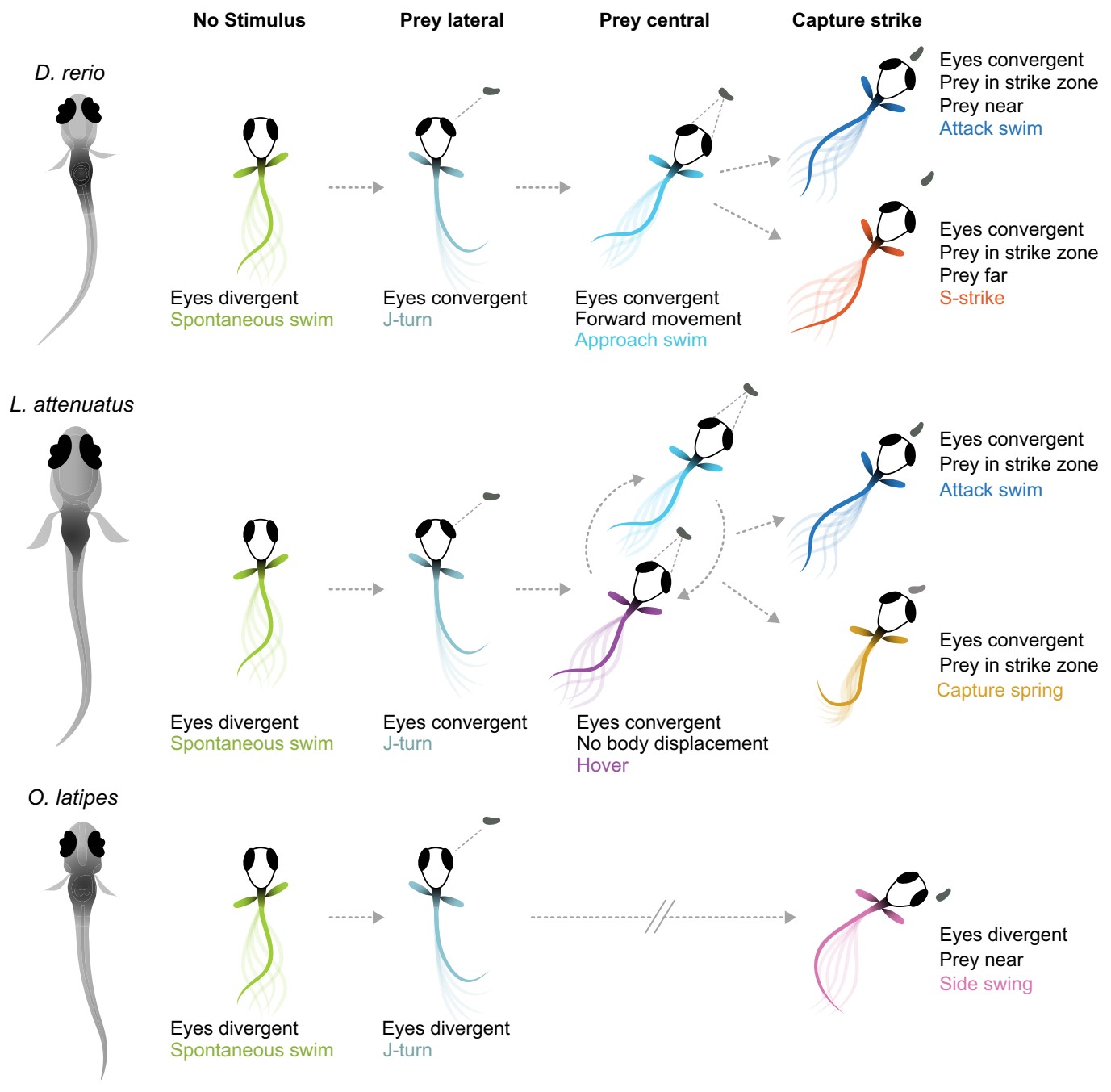

**Figure 6.** Schematic of hunting strategies in teleost larvae. Top: prey capture in zebrafish larvae (*D. rerio*) begins with eye convergence and a J-turn to orient toward the prey. The prey is then approached with a series of low-amplitude approach swims. Once in range in the central visual field, the prey is captured with an attack swim or an S-strike. Middle: prey capture in cichlids (represented by *L. attenuatus*) also begins with eye convergence and a J-turn. Prey is approached with a wide variety of tail movements. The prey is captured when it is in the central visual field with either an attack swim or a capture spring, during which the tail coils over several hundreds of milliseconds. Bottom: prey capture in medaka (*O. latipes*) begins with reorienting J-like turns, but these do not centralize prey in the visual field. Instead, the prey is kept lateral in the visual field and is captured with a side swing.

intermittent pauses, while medaka swim continuously with longer bouts ('gliding'). We speculate that the pauses may serve to sample the distance to the prey by comparing its position across the two eyes. In contrast, to sample motion cues, it is advantageous to perform continuous glides and swim lateral to the prey up until it is close enough for a sideways strike. Thus, natural selection may have favored specific locomotor (bouts vs. glides) and oculomotor (convergent vs. divergent) adaptations

of behavioral control depending on the dominant distance measurement mechanism (binocular vs. monocular) (*Figure 6*). While the behavioral evidence for this conjunction is still circumstantial and calls for further comparative work, this scenario makes testable predictions about its neurobiological and genetic implementation. Cichlids and medaka may also use size, brightness, and contrast cues to estimate prey distance in addition to the binocular disparity and monocular parallax mechanisms we have postulated here.

We have demonstrated that eye convergence is not always a hallmark of prey capture in fish larvae, and that multiple, kinematically distinct capture swims exist at these early stages in different species. We speculate that ocular vergence angles and swim kinematics for prey capture might coevolve, with S-strike-like captures and attack swims being associated with convergent eyes and side-swing captures occurring in the absence of eye convergence. These two strategies may have been present in a common ancestor of teleosts, with one or the other becoming dominant in certain lineages. Comparing neural circuitry across these (now) experimentally tractable animals could help pinpoint evolvable circuit nodes ('hotspots' of evolutionary change; *Roberts et al., 2022*; *Seeholzer et al., 2018*) and genetic loci that underlie prey localization strategies and swim kinematics in fishes.

# Materials and methods

**Key resources table**

| Reagent type (species) or resource | Designation | Source or reference | Identifiers | Additional information |
|---|---|---|---|---|
| Biological sample (*Lamprologus ocellatus*) | LO | Other | | From Alex Jordan, Max Planck Institute of Animal Behavior, Konstanz |
| Biological sample (*Neolamprologus multifasciatus*) | NM | *Bose et al., 2020* | | From Alex Jordan, Max Planck Institute of Animal Behavior, Konstanz |
| Biological sample (*Lepidiolamprologus attenuatus*) | LA | Other | | From Alex Jordan, Max Planck Institute of Animal Behavior, Konstanz |
| Biological sample (*Astatotilapia burtoni*) | AB | Other | | From Alex Jordan, Max Planck Institute of Animal Behavior, Konstanz |
| Biological sample (*Oryzias latipes* –wild-type) | Medaka; OL | Other | | From Joachim Wittbrodt, University of Heidelberg |
| Software | SLEAP | *Pereira et al., 2019*; *Pereira et al., 2022* | | Social LEAP Estimates Animal Pose |
| Software | YOLO | *Redmon et al., 2016* | | You Only Look Once |
| Software, algorithm | Python code | *Mearns et al., 2020* | | Python code |
| Software, algorithm | Custom analysis code (Python 3) | GitHub, copy archived at *Mearns, 2025* | | Custom analysis code (Python 3) |

## Species used and animal experiments

The following species were used in this study: LO (n=14), LA (n=12), NM (n=22), and AB (n=14); medaka, OL (n=3). Adult fish were raised at the Max Planck Institute for Biological Intelligence. The cichlid colonies were originally provided by the Max Planck Institute of Animal Behavior, Konstanz, Germany. LO were housed in 50 l tanks with one male and two females and a shell for each individual for egg laying. LA were housed in groups (10–15 mixed-sex individuals) within a 50 l tank with a half flower pot for egg laying. NM were housed in large groups (20–30 mixed-sex individuals) within a 160 l tank with shells for egg laying. AB were housed in 160 l tanks in groups of 15 (1 male and 14 females) with plants to encourage courtship arenas. OL were raised like zebrafish. All behavioral observations were performed in accordance with applicable laws and approved by the Regierung Oberbayern (ROB AZ 55.2.1.54-2532-101-12, ROB-55.2Vet-2532.Vet_02-16-31, ROB-55.2–2532.Vet_02-19-16 and ROB-55.2–2532.Vet_02-22-59).

Adult cichlids were fed live artemia, kept under a 13 hr:11 hr day-night cycle in 27°C water (pH 8.2, conductivity ~550 μS) on sand substrate. Egg-laying behavior was monitored, and clutches were collected at 7–8 dpf from the shell (LO, NM), the flower pot half (LA), or gently released from the mouth of the mother (AB). OL embryos were harvested from natural crosses. Clutches were incubated at 28°C in sterile Petri dishes with filtered fish facility water. Larvae were assayed as soon as they

displayed robust feeding behaviors: LO and LA at 12 dpf, NM at 14 dpf, AB at 14–16 dpf, and OL at 10 dpf.

## Video acquisition

High-speed (300–500 frames/s) videography data of single larvae of each species were collected as described previously (*Mearns et al., 2020*). Behavior chambers were molded out of 2% agarose and measured 30 mm × 30 mm for cichlids or 20 mm × 20 mm for medaka. Single larvae were introduced into chambers with a drop of food culture (*Artemia salina* for cichlids, ~30 each; *Paramecium multimicronucleatum* [Carolina Biological Supply Company, Burlington, NC, USA] for medaka). Each animal was recorded for 10–15 min using a high-speed camera (EoSens CL MC1362, Mikrotron; objective: Sigma 50 mm f/2.8 ex DG Macro, Japan), backlit with a custom-built infrared LED array.

## Tracking

To track the tail and eyes, we trained neural networks using SLEAP (*Pereira et al., 2022*). We designed a 12-point skeleton to track the tail (left eye center, right eye center, rostral tip, swim bladder, and eight points along the tail), and a 7-point skeleton to track the eyes (swim bladder, and a nasal, central, and temporal point on the perimeter of each eye).

### Tail tracking

For tail tracking, a single instance model was trained for LA on a total of 701 labeled frames from eight videos across eight fish. 290 frames were manually labeled, and 411 were corrected across two rounds of human-in-the-loop training. This model was used to track the tail in LO and NM in addition to LA. To track AB and medaka, the LA base model was trained with an additional 345 and 190 frames, respectively, from each of these species.

### Eye tracking

For the eyes, we first tracked 4900 frames of LA behavior using an unsupervised tracking algorithm. These points were used to train a base model. This base model was further refined for each species using human-in-the-loop on an additional 402 frames, AB; 331 frames, OL; and 314 frames, LO (this model was also used to track the eyes in NM).

### Prey tracking

Artemia were tracked using a single neural network developed for object detection, YOLO (*Redmon et al., 2016*). We trained the model on 128 frames. Due to the lower prey contrast in the AB and medaka videos, we could not track prey location in these videos. Prey locations prior to the medaka swing were manually annotated in 17 frames where the targeted paramecium was clearly visible.

To obtain continuous tracks of the most likely targeted prey, we tracked prey location across all frames of episodes of eye convergence. We tracked identities to single artemia across frames using the Hungarian algorithm. We assumed that the artemia nearest to the larva in the anterior visual field in the 100 ms leading up to eye deconvergence was the targeted prey.

## Statistical analysis

All analysis was performed using custom-written Python code (libraries: numpy, pandas, scikit-learn, scipy, opencv).

We used $\alpha=0.05$ with a Bonferroni correction when assessing significance. In *Figures 2 and 4*, we used the Mann-Whitney U test from the scipy.stats module. For bootstrap tests in *Figures 1 and 3*, we compared the difference in median between groups and computed the probability of observing this value or greater under the null hypothesis. To build distributions under the null hypothesis, we recomputed the test statistic from median-adjusted data (sampling 10% of the data with replacement) 100,000 times.

Comparing statistically significant differences between groups in *Figures 1 and 3*: If two groups share a common letter (a, b, c, etc.), they are not significantly different (p-value > 0.05). If two groups do not share a common letter then they are significantly different (p-value ≤ 0.05).

**Eye convergence analysis**

## Computing eye convergence angles

The angle of each eye in each frame was calculated as the angle between the heading vector of the fish (vector between the midpoint of the fins and midpoint of the two eyes) and the long axis of the eye (vector between the temporal and nasal points). Zero signifies the long axis of the eye is parallel with the heading vector of the fish; positive angles signify nasalward rotation. The convergence angle was computed as the sum of these two angles (positive convergence indicates both eyes are rotated inward toward the midline). Eye angles of <−10° (diverged) or >60° converged indicated tracking errors and were excluded from further analysis.

## Gaussian mixture modeling

To determine when the eyes were converged, we fitted a Gaussian mixture model to the distribution of convergence angles for each species. We tested models with either one or two mixtures to determine whether any discernible eye convergence was present. We used a Bayesian inference criterion to determine whether one or two mixtures resulted in a better fit. For all cichlid species, a two-mixture model outperformed a single Gaussian distribution. Convergence phases coincided with hunting events. For medaka (OL), a two-mixture model performed marginally worse than the single distribution model.

## Identification of hunting episodes

To identify periods of hunting in cichlids, we computed the probability of eye convergence in each frame as the probability it belonged to the Gaussian mixture with the higher mean. This yielded a time series of eye convergence probability, which we smoothed with a 200 ms rolling mean. We defined a hunting episode as any continuous period where the probability of eye convergence was above 50%, with the additional condition that that convergence probability was above 80% for at least half this period.

**Tail kinematic analysis**

We tracked eight equally spaced points from the tail base to the tail tip in each frame using the whole-body SLEAP model (see above). In each frame, we represented the shape of a tail as a seven-dimensional vector, computed by calculating the angle between a line through each consecutive pair of tail points and the heading vector (from the midpoint of the fins to the midpoint between the eyes).

## PCA and data curation

To reduce the dimensionality of pose data, we performed PCA as described previously (*Mearns et al., 2020*). We concatenated the seven-dimensional tail vectors in time (all animals pooled, grouped by species, or for all species combined). We z-scored the data matrix prior to PCA by subtracting the mean angle at each point and dividing by the standard deviation. This ensures equal weighting of all points of the tail during PCA, since otherwise loadings in the PCs would be biased toward points closer to the tail tip, potentially missing subtle but important curvature in the rostral trunk.

PCA provides a set of PCs (orthogonal vectors, representing some loading across the input dimensions), each associated with an eigenvalue (variance in the data captured by that component). Each PC can be represented as an 'eigenfish' by mapping the raw loadings multiplied by some scalar back into the original space (*Mearns et al., 2020*). In all cases, we found that three components were sufficient to capture over 90% of the variance in the data, and so for subsequent analyses, we represented tail kinematics by a time-varying three-dimensional vector representing the projection of the tail shape onto the first three PCs in each frame.

Since we were unable to track every tail point in every frame of our data, we performed some interpolation to fill in missing frames in our tracking. For any period where there was a gap of four or fewer frames (≤10 ms) of missing tracking data, we inferred the tail shape using a cubic spline interpolation in each PC dimension. Gaps greater than four frames were excluded from our analyses. We found that tracking errors were easiest to detect by inspecting the weight of the third PC in each frame. Therefore, we excluded frames where the absolute value of this PC exceeded 10 standard deviations (or 5,

in the case of AB). We linearly interpolated tail kinematics so that all trials had a common sample rate in cases where the frame rates of videos did not match.

## Segmentation

We adapted our previously published bout detection algorithm for zebrafish larvae (*Mearns et al., 2020*) to segment the more continuous swimming behavior of the larval species used in this study. This algorithm involves five steps:

1. The tail is represented as a single-dimensional time series by computing the L2 norm in PC space for each frame.
2. The absolute value of the derivative of this trace was convolved with a Gaussian kernel (standard deviation: 40 ms). This produces a smooth trace with peaks whenever the shape of the tail is changing rapidly in a noncyclic manner (i.e. at transitions between behaviors).
3. A threshold was automatically computed by performing a kernel density estimation across all values of this trace (bandwidths: 0.005, NM, AB, and OL; 0.01, LA; 0.02, LO). If this generated a bimodal distribution, we set the threshold to the antimode (local minimum) of this distribution; otherwise, we set the threshold to the first inflection point after the mode.
4. Peaks in the traces (step 2) that were above the threshold (step 3) were detected (scipy.signal. find_peaks function). Behavioral syllables ('bouts') were segmented at local minima between these peaks.
5. We performed a round of cleaning to exclude noisy bouts. For each bout, we computed the standard deviation in each PC dimension (low standard deviations in every dimension suggest no tail movement but rather noise in tracking). We then fit a Gaussian mixture model to the three-dimensional space of these standard deviations and excluded bouts predicted to belong to the mixture with the smallest standard deviation across PC dimensions. For each species, we manually set the number of mixture components to exclude most noisy bouts while keeping real movement in the dataset.

This pipeline identified the following numbers of bouts in each species: 2067 (LO); 11,237 (NM); 10,445 (LA); 9796 (AB); 3437 (OL).

## Alignment

Having segmented behavior into individual bouts, we sought to align these bouts for clustering and embedding. Our segmentation algorithm produced bouts of varying duration; however, we reasoned that the most important features of a bout that would distinguish it from others would occur when the tail is in the 'most extreme' or characteristic position, so we aligned all bouts to their peak distance in PC space (see segmentation step 1). Then, we truncated or zero-padded the aligned bouts such that we fully encompassed 80% of bout starts (before the peak) and 80% of bout ends (after the peak).

To aid the identification of behavioral clusters, we chose to ignore the left/right orientation of bouts. Therefore, we flipped bouts such that at their peaks, the tail was always oriented in the same direction. We performed an initial clustering step using Ward hierarchical clustering with 30 clusters. For this clustering step, we included each bout and its mirror image to produce symmetric clusters. To determine the directionality of a bout, we computed the mean value of the first PC at the peak of each cluster and used either the original or mirrored version of the bout accordingly. We found flipping bouts based on cluster was more robust than flipping each bout individually, since in many instances the value of the first PC is close to zero at the bout peak, and we wanted to ensure all similar bouts were oriented in the same direction to aid embedding and subsequent clustering.

## Clustering

To identify behavioral clusters, we performed affinity propagation on the aligned, flipped kinematic motifs. We treated each motif as a single vector, concatenating the time series for each PC. Affinity propagation computes a set of clusters (represented by 'exemplar' data points) for a given dataset at a given preference value. Lower preference values produce fewer clusters, so we used the minimum similarity between data points, which is a standard metric for computing a preference value (*Frey and Dueck, 2007*). This approach yielded between 11 and 32 clusters per species (11, LO; 23, NM; 29, LA; 32, AB; 17, OL), which is a reasonable number to expect in these data, being the same order as the number of behaviors that have been identified in zebrafish larvae (*Marques et al., 2018*).

## Embedding

To visualize behavior maps, we embedded motifs in a two-dimensional space using t-SNE (*Maaten and Hinton, 2008*), concatenating the time series for each PC such that each motif was represented by a single vector.

## Prey capture bouts

To assign bouts a prey capture score, we computed the probability of eye convergence (see *Eye convergence analysis*, above) at the peak of each bout. We then averaged these probabilities over all bouts belonging to a given cluster to compute the prey capture score for that cluster. Thus, a prey capture score of zero signifies zero probability of eye convergence for all bouts within a cluster, and a prey capture score of one indicates a 100% probability of eye convergence for all bouts within a cluster.

## Capture strike analysis

### Identifying putative strikes

To identify putative capture strikes in cichlids, we found all bouts whose peak coincided with the end of a hunting episode (see *Eye convergence analysis*, above), allowing the peak to occur up to 100 ms before and up to 200 ms after the detected end of eye convergence. Since we could not use eye convergence to detect prey capture in medaka, we manually inspected clusters obtained from analyzing tail movements and found that one of these clusters corresponded to a strike behavior (side swing). For cross-species comparisons, all data were interpolated to a common frame rate of 400 fps. We performed an initial round of clustering on these putative strike bouts (Ward hierarchical clustering with n clusters = 8) and removed small clusters containing fewer than 20 bouts. After this filtering, we had 341 putative strike bouts (16, LO; 31, NM; 191, LA; 59, AB; 44, OL).

### Clustering strikes

To only consider tail kinematics around the moment of the strike, we truncated bouts to a 150-frame window (375 ms) around the peak. We then clustered these bouts with Ward hierarchical clustering, with the number of clusters set to five (same as the number of species). To determine whether each species had its own unique strike behavior or whether the same behaviors were shared across species, we computed the confusion matrix using the cluster labels and species identity labels. Two clusters corresponded to abort behaviors in cichlids, two clusters corresponded to cichlid capture strikes (attack swim and capture spring), and one cluster corresponded to medaka side swings.

### Change in heading during prey capture

To compute the change in heading leading up to a strike, we considered a 200 ms window before the peak of each strike bout. So as to not bias our analysis to the rapid changes in heading at the moment of the medaka side swing, we only analyzed heading changes up to 25 ms before the peak of each bout. We virtually rotated and mirrored all capture sequences such that the heading 25 ms before the bout peak was zero, and the mean heading prior to this moment was positive. For each event, we then computed the slope of a regression line (least squares fit).

## Acknowledgements

We thank Alex Jordan for sharing cichlid stocks and Joachim Wittbrodt for sharing medaka stocks; Jessica Zung, Qing Wang, and Greg Marquart for critical feedback on the manuscript; Abdelrahmen Adel for technical assistance training neural networks; Krasimir Slanchev and the animal caretakers at the Max Planck Institute for Biological Intelligence for assistance raising and breeding fish; and Julia Kuhl for illustrations. Funding was provided by the Max Planck Society. MWS was supported by a Boehringer Ingelheim Fonds Fellowship.

# Additional information

### Funding

| Funder | Grant reference number | Author |
| --- | --- | --- |
| Max Planck Society | | Sydney A Hunt<br>Martin W Schneider<br>Ash V Parker<br>Manuel Stemmer<br>Herwig Baier |
| Boehringer Ingelheim Fonds | Fellowship | Martin W Schneider |

The funders had no role in study design, data collection and interpretation, or the decision to submit the work for publication. Open access funding provided by Max Planck Society.

### Author contributions
Duncan S Mearns, Sydney A Hunt, Conceptualization, Resources, Software, Formal analysis, Validation, Investigation, Visualization, Methodology, Writing – original draft, Writing – review and editing; Martin W Schneider, Methodology; Ash V Parker, Resources, Methodology; Manuel Stemmer, Resources; Herwig Baier, Conceptualization, Resources, Supervision, Funding acquisition, Validation, Investigation, Visualization, Methodology, Writing – original draft, Project administration, Writing – review and editing

### Author ORCIDs
Herwig Baier ⓘD https://orcid.org/0000-0002-7268-0469

### Ethics
All behavioral observations were performed in accordance with applicable laws and approved by the Regierung Oberbayern (ROB AZ 55.2.1.54-2532-101-12, ROB-55.2Vet-2532.Vet_02-16-31, ROB-55.2-2532.Vet_02-19-16, ROB-55.2-2532.Vet_02-22-59).

Reviewer #3 (Public review): https://doi.org/10.7554/eLife.98347.3.sa1
Author response https://doi.org/10.7554/eLife.98347.3.sa2

---

# Additional files

### Supplementary files
MDAR checklist

### Data availability
Datasets are available here: Mendeley Data https://doi.org/10.17632/m6zs9bm7w3.1.

The following dataset was generated:

| Author(s) | Year | Dataset title | Dataset URL | Database and Identifier |
| --- | --- | --- | --- | --- |
| Mearns DS, Hunt SA, Baier H | 2024 | Diverse prey capture strategies in teleost larvae | https://doi.org/10.17632/m6zs9bm7w3.1 | Mendeley Data, 10.17632/m6zs9bm7w3.1 |

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
