## [Editor Report · eLife Assessment]

This **important** body of work uses state-of-the-art quantitative methods to characterize and compare behaviors across five different fish species to understand which features are conserved and which ones are differentiated. The **convincing** results from this study will be of interest to ethologists and also have potential utility in understanding the neural mechanisms leading to these behaviors.

---

## [Referee Report · Reviewer #3 (Public review)]

Summary:

This paper uses 2D pose estimation and quantitative behavioral analyses to compare patterns of prey capture behavior used by six species of freshwater larval fish, including zebrafish, medaka, and four cichlids. The convincing comparison of tail and eye kinematics during hunts reveals that cichlids and zebrafish use binocular vision and similar hunting strategies, but that cichlids make use of an expanded set of action types. The authors also provide convincing evidence that medaka instead use monocular vision during hunts. This finding has important implications for the evolution of distinct distance estimation algorithms used by larval teleost fish species during prey capture.

Strengths:

The quality of the behavioral data is solid and the high frame rate allowed for careful quantification and comparison of eye and tail dynamics during hunts. The statistical approach to assess eye vergence states (Figure 2B) is elegant, the cross-species comparison of prey location throughout each hunt phase is well done (Figure 3B-D), and the demonstration that swim bout tail kinematics from diverse species can be embedded in a shared "canonical" principal component space to explain most of the variance in 2D postural dynamics for each species (Figure 4A-C) provides a simple and powerful framework for future studies of behavioral diversification across fish species.

---

## [Author Response]

The following is the authors’ response to the original reviews.

**Reviewer 1:**
(1) The overall conclusion, as summarized in the abstract as "Together, our study documents the diversification of locomotor and oculomotor adaptations among hunting teleost larvae" is not that compelling. What would be much more interesting would be to directly relate these differences to different ecological niches (e.g. different types of natural prey, visual scene conditions, height in water column etc), and/or differences in neural circuit mechanisms. While I appreciate that this paper provides a first step on this path, by itself it seems on the verge of stamp collecting, i.e. collecting and cataloging observations without a clear, overarching hypothesis or theoretical framework.

There are limited studies on the prey capture behaviors of larval fishes, and ours is the first to compare multiple species systematically using a common analysis framework. Our analysis approach could have uncovered a common set of swim kinematics and capture strategies shared by all species; but instead, we found that medaka used a monocular strategy rather than the binocular strategy of cichlids and zebrafish. Our analysis similarly could have revealed first-feeding larvae of all species go through a “bout” stage, which was previously proposed as important for sensorimotor decision making (Bahl et al., 2019), but instead we found that medaka and some cichlids have more continuous swimming from an early life stage. Finally, the rate at which prey capture kinematics evolves is not known. Our approach could have revealed rapid diversification of feeding strategies in cichlids (similarly to how adult feeding behavior evolves), but instead we found smaller differences within cichlids than between cichlids and medaka.

(2) The data to support some of the claims is either weak or lacking entirely.

Highlighted timestamps in videos, new stats in fig 1H and fig 2, updated supplementary figures now provide additional support for claims.

- It would be helpful to include previously published data from zebrafish for comparison.

We appreciate the suggestion. Mearns et al. (2020) provided a comprehensive account of prey capture in zebrafish larvae in an almost identical setup with similar analyses. We do not feel it is necessary to recount all the findings in that paper here. There are many studies on prey capture in zebrafish from the past 20 years, and reproducing these here would not add anything to that extensive pre-existing literature.

- Justification is required for why it is meaningful to compare hunting strategies when both fish species and prey species are being varied. For instance, artemia and paramecia are different sizes and have different movement statistics.

We added text explaining why different food was chosen for medaka/cichlids. There is no easy way to stage match fishes as evolutionarily diverged as cichlids, medaka, and zebrafish. Size is a reasonable metric within a species, but there is no guarantee that sizematched larvae of two different species are at the same level of maturity. Therefore, we thought the most appropriate stage to address is when larvae first start feeding, as this enables us to study innate prey capture behavior before any learning or experience-dependent changes have taken place. Given that zebrafish, medaka and cichlid larvae are different sizes when they first start feeding, it was necessary to study their hunting behavior to different prey items.

- It would be helpful in Figure 1A to add the abbreviations used elsewhere in the paper. I found it slightly distracting that the authors switch back and forth in the paper between using "OL" and "medaka" to refer to the same species: please pick one and then remain consistent.

Medaka is the common name for the japanese rice fish, O. latipes. Cichlilds do not have common names are only referred to by their scientific names. Since readers are more likely to be familiar with the common name, medaka, we now use medaka (OL) throughout the manuscript, which we hope makes the text clearer.

- The conceptual meaning of behavioral segmentation is somewhat unclear. For zebrafish, the bouts already come temporally segmented. However in medaka for instance, swimming is more continuous, and the segmentation is presumably more in terms of "behavioral syllables" as have been discussed for example mouse or *Drosophila* behavior (in the last row of Figure S1 it is not at all obvious why some of the boundaries were placed at their specific locations). It's not clear whether it's meaningful to make an equivalence between syllables and bouts, and so whether for instance Figure 1H is making an apples-to-apples comparison.

We clarified the text to say we are comparing syllables, rather than bouts.

- The interpretation of 1H is that "medaka exhibited significantly longer swims than cichlids"; however this is not supported by the appropriate statistical test. The KS test only says that two probability distributions are different; to say that one quantity is larger than another requires a comparison of means.

Updated Fig 1H; boostrap test (difference of medians) and re plotted data as violin plots.

(2) The data to support some of the claims is either weak or lacking entirely.

Highlighted timestamps in videos, new stats in fig 1H and fig 2, updated supplementary figures now provide additional support for claims.

- I think the evidence that there are qualitatively different patterns of eye convergence between species is weak. In Figure 2A I admire the authors addressing this using BIC, and the distributions are clearly separated in LA (the Hartigan dip test could be a useful additional test here). However for LO, NM, and AB the distributions only have one peak, and it's therefore unclear why it's better to fit them with two Gaussians rather than e.g. a gamma distribution. Indeed the latter has fewer parameters than a two-gaussian model, so it would be worthwhile to use BIC to make that comparison. The positions of the two Gaussians for LO, NM, and AB are separated by only a handful of degrees (cf LA, where the separation is ~20 degrees), which further supports the idea that there aren't really two qualitatively different convergence states here.

Added explanation to text.

- Figure S2 is unfortunately misleading in this regard. I don't claim the authors aimed to mislead, but they have made the well-known error of using colors with very different luminances in a plot where size matters (see e.g. https://www.r-project.org/conferences/DSC-2003/Proceedings/Ihaka.pdf).Thus, to the eye, it appears there's a big valley between the red and blue regions, but actually, that valley is full of points: it's really just one big continuous blob.

Kernel density estimation of eye convergence angles were added to Figure S2. The point we wish to make is that there is higher density when both eyes are rotated invwards (converged) in cichlids, but not medaka (O. latipes). The valley between converged and unconverged states being full of points is due to (1) slight variation with placement of key points in SLEAP, which blurs the boundary between states and (2) the eye convergence angle must pass through the valley in order to become converged, so necessarily there are points in between the two extremes of eye convergence.

- In Figure 2D please could the authors double-check the significance of the difference between LO and NM: they certainly don't look different in the plot.

Thank for for flagging this. We realize the way we previously reported the stats was open to misinterpretation. We have updated figure 2C, D and F to use letters to indicate statistical groupings, which hopefully makes it clearer which species are statistically different from each other.

- In Figure 2G it's not clear why AB is not included. It is mentioned that the artemia was hard to track in the AB videos, but the supplementary videos provided do not support this.

The contrast of the artemia in the AB videos is sufficiently different from the other cichlid videos that our pre-trained YOLO model fails. Retraining the model would be a lot of extra work and we feel like a comparison of three species is sufficient to address the sensorimotor transformations that occur over the course of prey capture in cichlids.

- The statement "Zebrafish larvae have a unique swim repertoire during prey capture, which is distinct from exploratory swim bouts" is not supported by the work of others or indeed the authors' own work. In Figure 4F all types of bouts can occur at any time, it's just the probability at which they occur that varies during prey capture versus other times (see also Mearns et al (2020) Figure S4B).

The point is well taken that there probably is not a hard separation between spontaneous and prey capture swims based on tail kinematics alone, which is also shown in Marques et al. (2018). However, we think that figure 2I of Mearns et al., which plots the probability of swims being drawn from different parts of the behavior space during prey capture (eyes converged) or not (eyes unconverged), shows that the repertoire of swims during the two states is substantially different. Points are blue or red; there are very few pale blue/pale red points in that figure panel. Figure S4B is showing clustered data, and clustering is a notoriously challenging problem for which there exists no perfect solution (Kleinberg, 2002). The clusters in Mearns et al. incorporated information about transition structure, as this was necessary for obtaining interpretable clusters for subsequent analyses. However, a different clustering approach could have yielded different boundaries, which may have shown more (or less) separation of bout types during prey capture/exploratory swimming. Therefore, we have updated the text to say that zebrafish perferentially perform different swim types during prey capture and exploration, and re-interpreted the behavior of cichlids similarly.

- More discussion is warranted of the large variation in the number of behavioral clusters found between species (11-32). First, how much is this variation really to be trusted? I appreciate the affinity propogation parameters were the same in all cases, but what parameters "make sense" is somewhat dependent on the particular data set. Second, if one does believe this represents real variation, then why? This is really the key question, and it's unsatisfying to merely document it without trying to interpret it.

Extended paragraph with more interpretation.

- What is the purpose of "hovers"? Why not stay motionless? Could it be a way of reducing the latency of a subsequent movement? Is this an example of the scallop theorem?

Added a couple of sentences speculating on function.

- I'm not sure "spring-loaded" is a good term here: the tension force of a coiled tail is fairly negligible since there's little internal force actively trying to straighten it.

Rewrote this part to highlight that fish spring toward the prey, without the implication that tension forces in the tail are responible for the movement. However, we are not aware of any literature measuring passive forces within the tail of fishes. Presumably the notochord is relatively stiff and may provide an internal force trying to straighten the tail.

- There are now several statements for which no direct evidence is presented. We shouldn't have to rely on the author's qualitative impressions of what they observed: show us quantitative analysis.* "often hover"* "cichlids often alternate between approaches and hover swims"* "over many hundreds of milliseconds"* "we have also observed suction captures and ram-like attacks"* "may swim backwards"* "may expel prey from their mouth"* "cichlid captures often occur in two phases"

Added references to supplementary videos with timestamps to highlight these behaviors.

- I don't find it plausible that sated fish continue hunting prey that they know they're not going to eat just for the practice.

Removed the speculation.

- In Figure 3 is it not possible to include medaka, based on the hand-tracked paramecia?

The videos are recorded at high frame rate, so it would be a lot of additional work to track these manually. Furthermore, earlier in prey capture it is very difficult to tell by watching videos which prey the medaka are tracking, especially as single paramecia can drift in and out of focus in the videos. Since there is no eye convergence, it is very difficult to ascertain for certain when tracking a given prey begins. In Fig 4, it was only possible to track paramecia by hand since it is immediately prior to the strike and from the video it is possible to see which paramecium the fish targeted. Our analyses of heading changes was performed over the 200 ms prior to a strike, which we think is a conservative enough cutoff to say that fish were probably pursuing prey in this window (it is shorter than the average behavioral syllable duration in medaka).

- Figure 3 (particularly 3D) suggests the interesting finding that LA essentially only hunt prey that is directly in front of them (unlike LO and NM, the distribution of prey azimuth actually seems to broaden slightly over the duration of hunting events).This is worthy of discussion.

We offer a suggestion for the many instances of prey capture being initiated in the central visual field in LA later in the manuscript when we discuss spitting behavior. We have added text to make this point earlier in the manuscript. The increase in azimuthal range at the end of prey capture may be due to abort swims (e.g. supp. vid. 1, 00:21). The widening of azimuthal angles is present in LO and NM also and is not unique to LA.

- The reference Ding et al (2016) is not in the reference list.

Wrong paper was referenced. Should be Ding 2019, which has been added to bibliography.

- I am not convinced that medaka exhibit a unique side-swing behavior. I agree there is this tendency in the example movie, however, the results of the quantification (Figure 4) are underwhelming. First, cluster 5 in 4K appears to include a proportion of cases from LA and AB. These proportions may be small, but anything above zero means this is not unique to medaka. Second, the heading angle (4N) starts at 4 degrees for LA and 8 degrees for medaka. This difference is genuine but very small, much smaller than what's drawn in the schematic (4M). I'm not sure it's justifiable to call a difference of 4 degrees a qualitatively different strategy.

We have changed the text to highlight that side swing is highly enriched in medaka. Comparing 4J to 3B we would argue that there is a qualitative difference in the strategy used to capture prey in the cichlid larvae we study here and medaka. We agree that further work is required to understand distance estimation behaviors in different species. In this manuscript, we use heading angle as a proxy for how prey position might change on the retina over a hunting sequence. But as the heading and distance are changing over time, the actual change in angle on the retina for prey may be much larger than the ~8 degree shift reported here. The actual position of the prey is also important here, which, for reasons mentioned above, we could not track. Given the final location of prey in the visual field prior to the strike (Fig 4J), the most parsimonious explanation of the data is that the prey is always in the monocular visual field. In cichlids, the prey is more-or-less centered in the 200 ms preceding the strike. While it is true theat the absolute difference in heading is 4 degrees, when converted to an angular velocity (4N, right), the medaka (OL) effectively rotate twice as fast as LA (20 deg/s vs 40 deg/s), which we think is a substantial difference and evidence of a different targeting strategy.

- 4K: This is referred to in the caption as a confusion matrix, which it's not.

Fixed.

- 4N right panel: how many fish contributed to the points shown?

Added to figure legend (n=113, LA; n=36, OL). Same data in left and right panels.

- In the Discussion it is hypothesized that medaka use their lateral line in hunting more than in other species. Testing this hypothesis (even just compared to one other species) would be fairly straightforward, and would add significant interest to the paper overall.

We agree that this is an interesting experiment for follow up studies, but it is beyond the scope of the current manuscript as we do not have the appropriate animal license for this experiment.

**Reviewer 2:**
The paper is rather descriptive in nature, although more context is provided in the discussion. Most figures are great, but I think the authors could add a couple of visual aids in certain places to explain how certain components were measured.

Added new supplemental figure (Supp Fig 2)

Figure 1B- it could be useful to add zebrafish and medaka to the scientific names (I realize it's already in Figure A but I found myself going back and forth a couple of times, mostly trying to confirm that O. latipes is medaka).

Added common names to 1B, sprinkled reminders of OL/medaka throughout text.

Figure 1G. I wasn't sure how to interpret the eye angle relative to the midline. Can they rotate their eyes or is this due to curvature in the 'upper' body of the fish? Adding a schematic figure or something like that could help a reader who is not familiar with these methods. Related to this, I was a bit confused by Figure 2A. After reading the methods section, I think I understand - but I little cartoon to describe this would help. It also reminds the reader (especially if they don't work with fish) that fish eyes can rotate. I also wanted to note that initially, I thought convergence was a measure of how the two eyes were positioned relative to the prey given the emphasis given on binocular vision, and only after reading certain sections again did I realize convergence was a measure of eye rotation/movement.

New supplemental figure explaining how eye tracking is performed

Figure 3. It was not immediately clear to me what onset, middle, and end represented - although it is explained in the caption. I think what tripped me up is the 'eye convergence' title in the top right corner of Figure 3A.

Updated figure with schematic illustrating that time is measured relative to eye convergence onset and end.

The result section about attack swim, S-strike, capture spring, etc. was a bit confusing to read and could benefit from a couple of concise descriptions of these behaviors. For example, I am not familiar with the S strike but a couple of paragraphs into this section, the reader learns more about the difference between S strike vs. attack swim. This can be mentioned in the first paragraph when these distinct behaviors are mentioned.

Added description of behavior earlier in text.

Figure 4. Presents lots of interesting data! I wonder if using Figure 1E could help the reader better understand how these measurements were taken.

New supplemental figure added, explaining how tail tracking is performed.

I probably overlooked this, but I wonder why so many panels are just focused on one species.

Added explanation to the text.

Is the S-shaped capture strategy the same as an S strike?

Clarified in text to say "S-strike-like". This is a description of prey capture from adult largemouth bass in New et al. (2002). From the still frames shown in that paper, the kinematics looks similar to an S-strike or capture spring. The important point we wish to make is that tail is coiled in an S-shape prior to a strike, which indicates this that a kinematically similar behavior exists fishes beyond just larval cichlids and zebrafish.

At the end of the page, when continuous swimming versus interrupted swimming is discussed, please remind the reader that medaka shows more continuous swimming (longer bouts).

Added "while medaka swim continuously with longer bouts ("gliding")".

After reading the discussion, it looks like many findings are unique. For example, given that medaka is such a popular model species in biology, it strikes me that nobody has ever looked into their hunting movements before. If their findings are novel, perhaps they should state so it is clear that the authors are not ignoring the literature.

We have highlighted what we believe to be the novelty of our findings (first description of prey capture in larval cichlids and medaka). To our knowledge, we are first to describe hunting in medaka; but there is an extensive literature on medaka dating back to the early 20th century, some of which is only published in Japanese. We have done our best to review the literature, but we cannot rule out that there are papers that we missed. No English language article or review we found mentions literature on hunting behavior in medaka larvae.

**Reviewer 3:**
More evidence is needed to assess the types of visual monocular depth cues used by medaka fish to estimate prey location, but that is beyond the scope of this compelling paper. For example, medaka may estimate depth through knowledge of expected prey size, accommodation, defocus blur, ocular parallax, and/or other possible algorithms to complement cues from motion parallax.

Added sentence to discussion highlighting that other cues may also contribute to distance estimation in cichlids and medakas. Follow-up studies will require new animal license.

None. It's quite nice, timely, and thorough work! For future work, one could use 3D pose estimation of eye and prey kinematics to assess the dynamics of the 2D image (prey and background) cast onto the retina. This sort of representation could be useful to infer which monocular depth cues may be used by medaka during hunting.

Great suggestion for follow up studies. Bolton et al. and Mearns et al. both find changes in z associated with prey capture, and it would be interesting to see how other fish species use the full 3-dimensional water column during prey capture, especially considering the diversity of hunting strategies in adult cichlids (ranging from piscivorous species, like LA, to algar grazers).

In Figure 4N, you use "change in heading leading up to a strike as a proxy for the change in visual angle of the prey for cichlids and medaka." This proxy makes sense, but you also have the eye angles and (in some cases) the prey positions. One could estimate the actual change in visual angle from this information, which would also allow one to measure whether the fish are trying to stabilize the position of the prey on a high-acuity patch of the retina during the final moments of the hunt. This information may also shed light on which monocular depth cues are used.

As addressed in comment to reviewer 1, this would require actually manually tracking individual paramecia over hundreds of frames. It is not possible to determine exactly when hunting begins in medaka, and it is prone to errors if medaka switch between targets over the course of a hunting episode. This question is better addressed with psychophysics experiments in embedded animals where it is possible to precisely control the stimulus, but this requires new animal licenses and is beyond the scope of this paper.

In Figure 5, you could place the prey object a little farther from the *D. rerio* fish for the S-strike diagram.

Fixed.

Figure 4F legend should read "...at the peak of each bout."

Fixed.